# Development and validation of a multimorbidity risk prediction nomogram among Chinese middle-aged and older adults: a retrospective cohort study

Xiao Zheng,[1,2] Benli Xue,[2,3] Shujuan Xiao,[2,3] Xinru Li,[2,3] Yimin Chen,[2,3] Lei Shi  ,[3] Xiaoyan Liang,[1,3] Feng Tian,[1,3] Chichen Zhang  [2,3]

For numbered affiliations see end of article.

**Correspondence to**
Professor Chichen Zhang;
zhangchichen@sina.com and
Professor Feng Tian;
qqtina@smu.edu.cn

## ABSTRACT

**Objectives** The aim of this study is to establish a self-simple-to-use nomogram to predict the risk of multimorbidity among middle-aged and older adults.

**Design** A retrospective cohort study.

**Participants** We used data from the Chinese Longitudinal Healthy Longevity Survey, including 7735 samples.

**Main outcome measures** Samples' demographic characteristics, modifiable lifestyles and depression were collected. Cox proportional hazard models and nomogram model were used to estimate the risk factors of multimorbidity.

**Results** A total of 3576 (46.2%) participants have multimorbidity. The result showed that age, female (HR 0.80, 95% CI 0.72 to 0.89), chronic disease (HR 2.59, 95% CI 2.38 to 2.82), sleep time (HR 0.78, 95% CI 0.72 to 0.85), regular physical activity (HR 0.88, 95% CI 0.81 to 0.95), drinking (HR 1.27 95% CI 1.16 to 1.39), smoking (HR 1.40, 95% CI 1.26 to 1.53), body mass index (HR 1.04, 95% CI 1.03 to 1.05) and depression (HR 1.02, 95% CI 1.01 to 1.03) were associated with multimorbidity. The C-index of nomogram models for derivation and validation sets were 0.70 (95% CI 0.69 to 0.71, p=0.006) and 0.71 (95% CI 0.70 to 0.73, p=0.008), respectively.

**Conclusions** We have crafted a user-friendly nomogram model for predicting multimorbidity risk among middle-aged and older adults. This model integrates readily available and routinely assessed risk factors, enabling the early identification of high-risk individuals and offering tailored preventive and intervention strategies.

## STRENGTHS AND LIMITATIONS OF THIS STUDY

⇒ The model was constructed based on behavioural and household-level risk factors.
⇒ We have used a culling method to deal with the missing values, this may lead to sample bias and the extrapolation of the model needs to be careful.
⇒ We could not compare the performance of the nomogram model with different models.

focus for public health interventions. Consequently, it is essential to understand the prevalence trends of multimorbidity and the contributing factors within populations. This understanding will enable individuals to estimate and modify their personal risk of developing multimorbidity.

Although age is undeniably a well-established risk factor for multimorbidity,[7] the multifaceted nature of this phenomenon demands a more comprehensive understanding that goes beyond age-related associations. Existing research has indeed confirmed the higher prevalence of multimorbidity among older adults, with systematic reviews reporting rates ranging from 55% to 98% in the elderly population.[8] Additionally, demographic factors such as female gender, and lower socioeconomic status have consistently been associated with an increased risk of multimorbidity. However, while these factors provide valuable insights, they represent only a fraction of the complex web of variables contributing to multimorbidity.

Previous studies have explored associations between socio-demographic factors, physical characteristics and social networks with multimorbidity.[9 10] For example, numerous studies have linked poor sleep quality to an elevated risk of multimorbidity.[11–14]

## BACKGROUND

Multimorbidity, commonly defined as the co-occurrence of two or more chronic conditions,[1 2] has emerged as a significant public health concern and poses challenges for healthcare systems. Extensive evidence[3] has shown that multimorbidity is associated with an increased risk of mortality,[4] reduced quality of life, heightened healthcare usage and elevated health costs.[5 6] Thus, the prevention of multimorbidity has become a crucial

Furthermore, the impact of body mass index (BMI) and smoking on multimorbidity prevalence underscores the intricate connections among these factors.[15] Additionally, the well-documented relationship between depression and common chronic diseases has been established,[16 17] with longitudinal cohort studies demonstrating bidirectional associations between depression and multimorbidity.[18]

Despite the substantial body of evidence regarding the associations among socio-demographic factors, social networks, lifestyle factors, depression and the risk of developing multimorbidity, there remains a notable gap in the literature. This gap pertains to the absence of a comprehensive multivariable prediction model that integrates all these factors, providing a holistic assessment of multimorbidity risk. Our study seeks to address this gap by developing and validating a novel risk assessment model that encompasses a broad spectrum of variables, including those mentioned above. Our aim is to equip individuals with a more accurate and personalised estimate of their risk of developing multimorbidity, contributing to a deeper understanding of this multifaceted health issue.

Wider determinants of health (WDHs) encompass a multitude of social, economic, political and environmental factors that exert influence on health outcomes across an individual's lifespan. This influential model of health determinants places constitutional factors such as sex, age and genetics at its core, surrounded by concentric layers that encompass individual lifestyle factors, followed by the broader determinants.[19] While the core attributes remain relatively fixed, the determinants become more modifiable as the layers extend outward. Existing research has identified that individual lifestyle factors significantly contribute to multimorbidity among older adults. Chudasama *et al* also found that adopting a healthier lifestyle was associated with longer life expectancy for middle-aged adults, regardless of the presence of multimorbidity.[4]

In the context of the WDHs framework, individual behaviours constitute the innermost layer, presenting opportunities for modification, particularly through self-health management.[20–22] The accurate assessment of one's risk of multimorbidity and the identification of potential risk factors represent critical initial steps in the journey of self-management. Therefore, the development of a user-friendly tool to assist individuals in estimating their risk of multimorbidity is of paramount significance.

A nomogram is a health risk appraisal model that offers individualised, evidence-based and highly accurate risk estimation.[23 24] It is easy to use and can facilitate self-management-related decision-making. It is a user-friendly tool that facilitates decision-making related to self-management. In this study, we have developed the first nomogram for predicting the risk of multimorbidity among middle-aged and older adults.

## METHODS
### Study population
We used data from the China Health and Retirement Longitudinal Study (CHARLS), a nationally representative survey of Chinese residents aged 45 years and above. The baseline survey was conducted in 2011 using a multistage probability sampling strategy and probability-proportional-to-size sampling technique to ensure national representativeness. Follow-up waves were conducted in 2013, 2015 and 2018. Detailed information regarding the purpose, design, sample and questionnaires used in CHARLS can be found in other studies.[6 25 26] For this study, participants below the age of 45 and those with missing values in any variables were excluded from the analysis. The selection process is outlined in online supplemental figure S1.

### Measurements
#### Multimorbidity
In this study, multimorbidity was defined as the presence of two or more chronic non-communicable diseases, whether physical or psychological.[6 27] We assessed multimorbidity by examining the presence of 14 specific non-communicable diseases. Physical chronic non-communicable diseases encompassed diagnosed conditions such as hypertension, dyslipidaemia, diabetes, cancer, chronic lung disease, liver disease, heart disease, stroke, kidney disease, digestive disease, asthma and arthritis. Psychological chronic non-communicable diseases included diagnosed emotional, nervous or psychiatric problems, as well as memory-related diseases (all diseases were self-reported and diagnosed chronic conditions). To identify individuals with multimorbidity, we calculated the number of chronic diseases present for each participant. The outcome was the time to multimorbidity.

#### The modifiable lifestyles
This study included four well-known healthy lifestyle factors[4]: physical activity (PA), smoking, alcohol consumption and diet behaviour. Besides, sleep and social activity were included in this study.

The physical activity questionnaire used in CHARLS closely resembled the short version of the International Physical Activity Questionnaire (IPAQ).[28] However, some differences existed between CHARLS and IPAQ, such as assessing PA for a 'usual week' instead of 'the last 7 days' and lacking information on sedentariness. Additionally, instead of continuous values, four discrete time durations ('< 30 min' '≥30 min' '< 4 hours' and '≥ 4 hours') were collected.[29] We calculated the median score for each intensity level and summed the number of different intensity levels using the metabolic equivalent (MET) as a reference. The weight of each intensity level was derived from the IPAQ scoring protocol. Detailed information on PA and its calculation process can be found in the study by Li *et al*.[26] The total weekly PA (MET-minutes/week) was calculated by multiplying the frequency, duration and

MET values. According to IPAQ, a minimum total PA of at least 600 MET-minutes/week was defined as regular PA, while <600 MET-minutes/week indicated a lack of regular PA.

Smoking was categorised as No (not current smoker) and Yes (current smoker) at the time of assessment. Alcohol consumption status was divided into two groups: No (Did not drink in the past 12 months or drinking frequency is less than weekly) and Yes (others). Regular eating behaviour was determined based on the frequency of meals per day, with having three meals on time considered as regular eating.

Based on studies conducted in developed countries, respondents' total sleep duration was classified into five categories: <6 hours, 6 to <7 hours, 7 to <8 hours, 8 to <9 hours and ≥9 hours.[30][31] According to the Healthy China initiative (2019–2030), the length of night-time sleep ≥7 hours was defined as enough sleep in this study.[32] Social activity was categorised as 'No' and 'Yes' based on engagement in social activities within the past 12 months.

### Demographic characteristics
Demographic characteristics included age, sex (male and female), marital status (others and married), residency (others and rural), education (primary education and below, secondary education and above) and BMI scores.

The covariates, including demographic characteristics and modifiable lifestyle factors, were gathered by baseline questionnaire.

### Depression
Depression was assessed using the Chinese version of the Center for Epidemiological Studies Depression scale (CES-D-10).[33] The CES-D-10 contains 10 items with four response options: rare, some days (1–2 days), occasionally (3–4 days) and most of the time (5–7 days).[25] The scales for each of the 10 items were adjusted to 0, 1, 2 and 3, resulting in a CES-D-10 score ranging from 0 to 30, with higher scores indicating more negative feelings during the past week.[34][35]

### Statistical analysis
The participants were randomly divided into a derivation set and a validation set at a ratio of 7:3. Participant characteristics, such as age and BMI, were summarised as mean±SD and counts with proportions for categorical features. Cox proportional hazard models were used to estimate the associations between modifiable lifestyles (including PA, smoking, alcohol consumption and diet behaviour), depression and other identified risk factors with the development of multimorbidity in middle-aged and older adults. HRs and 95% CIs were reported for the total population. Factors with a significant level of less than 0.05 in the univariable regression model were entered into the multivariable Cox proportional hazard model for adjustment.

A nomogram was developed based on the results of the multivariable cox proportional hazard model in

the derivation set. The nomogram assigns risk points to each variable by proportionally converting regression coefficients to a 0–100-point scale. The variable with the highest absolute value of the β coefficient is assigned 100 points. The risk points for other variables are calculated based on the ratio of risk points to the β coefficient of the highest variable. A Prognostic Index (PI) was calculated by summing the risk points corresponding to each weighted covariate. The nomogram was validated using the concordance index (C-index) calculated through 1000-fold bootstrap resampling to reduce overfit bias. The developed nomogram was then applied to the validation set. Model performance was further evaluated using a calibration curve, which superimposes both data sets for visual comparison of discrimination. All analyses were performed using R, V.3.0, p<0.05 was considered to indicate statistical significance.

### Patient and public involvement
The public were not involved in the design, or conduct, or reporting, or dissemination plans of this research.

## RESULTS
### Baseline characteristics
A total of 7735 participants were included in this study, with 5449 individuals in the derivation set and 2286 in the validation set. The baseline characteristics of the study sample are presented in table 1. The average age of participants in both data sets was 59.0±9.2 years, and in 2011, 3726 individuals (48.2%) had at least one chronic disease.

### Prevalence of multimorbidity
In 2018, a total of 3576 participants (46.2%) were found to have multimorbidity. Among these individuals, the prevalence of multimorbidity was higher among older adults compared with those under the age of 60 (51.0% vs 42.6%). Additionally, women exhibited a higher prevalence of multimorbidity compared with men (48.5% vs 43.8%). Moreover, married individuals (45.7% vs 48.6%) and those with higher education (42.1% vs 48.1%) had a lower prevalence of multimorbidity compared with others (figure 1).

### Predictor factors of multimorbidity
The results of the univariable Cox proportional hazard model for the entire data set (online supplemental table S1) indicated that several factors were significantly associated with an increased risk of multimorbidity. These factors included older age, being female (HR 1.16, 95% CI 1.08 to 1.24), having existing chronic disease (HR 2.85, 95% CI 2.66 to 3.06) and higher depression scores (HR 1.04, 95% CI 1.03 to 1.05). Conversely, a higher level of education (HR 0.83, 95% CI 0.77 to 0.90), marital status (HR 0.91, 95% CI 0.83 to 0.99), higher BMI scores (HR 1.04, 95% CI 1.03 to 1.05), smoking (HR 1.46, 95% CI 1.36 to 1.56), drinking (1.46, 95% CI 1.36 to 1.56), getting

**Table 1** Characteristics of the sample on the baseline-2011

| Characteristics | N (%) /S± $\overline{X}$ | | |
| --- | --- | --- | --- |
| | All | Derivation set | Validation set |
| Age (years) | | | |
| <60 | 4387 (56.7) | 3039 (55.8) | 1348 (59.0) |
| ≥60 | 3348 (43.3) | 2410 (44.2) | 938 (41.0) |
| Sex | | | |
| Male | 3744 (48.4) | 2684 (49.3) | 1060 (46.4) |
| Female | 3991 (51.6) | 2765 (50.7) | 1226 (53.6) |
| Hukou | | | |
| Others | 6423 (83.0) | 4504 (82.7) | 1919 (83.9) |
| Rural | 1312 (17.0) | 945 (17.3) | 367 (16.1) |
| Education | | | |
| Primary education and below | 5296 (68.5) | 3720 (68.3) | 1576 (68.9) |
| Secondary education and above | 2439 (31.5) | 1729 (31.7) | 710 (31.1) |
| Martial | | | |
| Others | 1304 (16.9) | 910 (16.7) | 394 (17.2) |
| Married | 6431 (83.1) | 4539 (83.3) | 1892 (82.8) |
| Chronic disease | | | |
| No | 4009 (51.8) | 2817 (51.7) | 1192 (52.1) |
| Yes | 3726 (48.2) | 2632 (48.3) | 1094 (47.9) |
| Sleep (hours) | | | |
| <7 | 3803 (49.2) | 2701 (49.6) | 1102 (48.2) |
| ≥7 | 3932 (50.8) | 2748 (50.4) | 1184 (51.8) |
| Social activity | | | |
| No | 3864 (50.0) | 2747 (50.4) | 1117 (48.9) |
| Yes | 3871 (50.0) | 2702 (49.6) | 1169 (51.1) |
| Regular physical activity | | | |
| No | 3422 (44.2) | 2407 (44.2) | 1015 (44.4) |
| Yes | 4313 (55.8) | 3042 (55.8) | 1271 (55.6) |
| Regular eating | | | |
| No | 1009 (13.0) | 707 (13.0) | 302 (13.2) |
| Yes | 6726 (87.0) | 4742 (87.0) | 1984 (86.8) |
| Smoke | | | |
| No | 3379 (43.7) | 2390 (43.9) | 989 (43.3) |
| Yes | 4356 (56.3) | 3059 (56.1) | 1297 (56.7) |
| Drink | | | |
| No | 3354 (43.4) | 2387 (43.8) | 967 (42.3) |
| Yes | 4381 (56.6) | 3062 (56.2) | 1319 (57.7) |
| Body mass index | 23.32±3.69 | 23.31±3.72 | 23.35±3.62 |
| Depression | 9.13±4.57 | 9.10±4.56 | 9.21±4.60 |

at least 7 hours of sleep (HR 0.70, 95% CI 0.66 to 0.75), engaging in regular PA (HR 0.83, 95% CI 0.77 to 0.88) were associated with a decreased risk of multimorbidity.

The multivariable Cox proportional hazard models incorporated 11 significant variables identified through univariate analysis. Variable selection was performed using the backward selection method. In comparison to individuals under the age of 50, those in the age groups of 50–59 (HR 1.19, 95% CI 1.05 to 1.34), 60–69 (HR 1.47, 95% CI 1.30 to 1.67) and 70–79 (HR 1.26, 95% CI 1.08 to 1.48) exhibit a higher risk of multimorbidity. Furthermore, being female (HR 0.80, 95% CI 0.72 to 0.89),

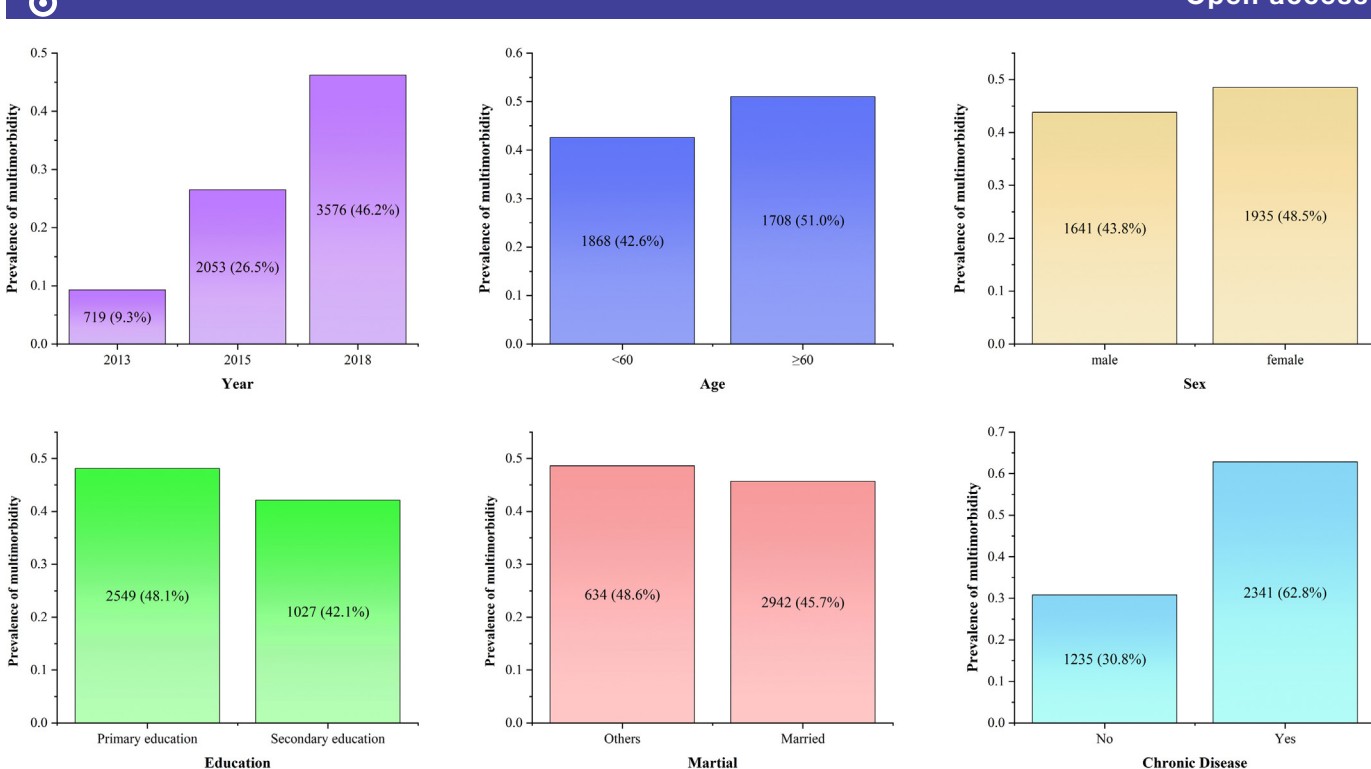

**Figure 1** Prevalence of multimorbidity in the population.

having a history of chronic disease (HR 2.59, 95% CI 2.38 to 2.82), getting at least 7 hours of sleep (HR 0.78, 95% CI 0.72 to 0.85), engaging in regular PA (HR 0.88, 95% CI 0.81 to 0.95), drinking (HR 1.27 95% CI 1.16 to 1.39), smoking (HR 1.40, 95% CI 1.26 to 1.53), higher BMI (HR 1.04, 95% CI 1.03 to 1.05) and increased depression scores (HR 1.02, 95% CI 1.01 to 1.03) were all found to be associated with multimorbidity in derivation set (table 2).

**Table 2** Factors associated with the risk of multimorbidity (multivariable cox proportional hazard model)

| Factors | Derivation set | | | Validation set | | |
|---|---|---|---|---|---|---|
| | HR | 95% CI | P value | HR | 95% CI | P value |
| Age (<50) | | | | | | |
| Age (50–59) | 1.19 | 1.05 to 1.34 | 0.005 | 1.20 | 1.01 to 1.43 | 0.049 |
| Age (60–69) | 1.47 | 1.30 to 1.67 | <0.001 | 1.26 | 1.04 to 1.52 | 0.018 |
| Age (70–79) | 1.26 | 1.08 to 1.48 | 0.037 | 1.24 | 0.99 to 1.56 | 0.060 |
| Age (≥80) | 0.95 | 0.72 to 1.25 | 0.725 | 1.01 | 0.66 to 1.55 | 0.964 |
| Sex (male) | | | | | | |
| Female | 0.80 | 0.72 to 0.89 | <0.001 | 0.71 | 0.60 to 0.84 | <0.001 |
| Chronic disease (No) | | | | | | |
| Yes | 2.59 | 2.38 to 2.82 | <0.001 | 2.70 | 2.37 to 3.06 | <0.001 |
| Sleep (<7 hours) | | | | | | |
| ≥7 hours | 0.78 | 0.72 to 0.85 | <0.001 | 0.74 | 0.65 to 0.83 | <0.001 |
| Regular physical activity (No) | | | | | | |
| Yes | 0.88 | 0.81 to 0.95 | 0.001 | 0.88 | 0.77 to 0.99 | 0.039 |
| Drink (No) | | | | | | |
| Yes | 1.27 | 1.16 to 1.39 | <0.001 | 1.35 | 1.17 to 1.56 | <0.001 |
| Smoke (No) | | | | | | |
| Yes | 1.40 | 1.26 to 1.53 | <0.001 | 1.57 | 1.34 to 1.84 | <0.001 |
| Body mass index | 1.04 | 1.03 to 1.05 | <0.001 | 1.02 | 1.01 to 1.03 | 0.001 |
| Depression | 1.02 | 1.01 to 1.03 | <0.001 | 1.02 | 1.01 to 1.04 | 0.001 |

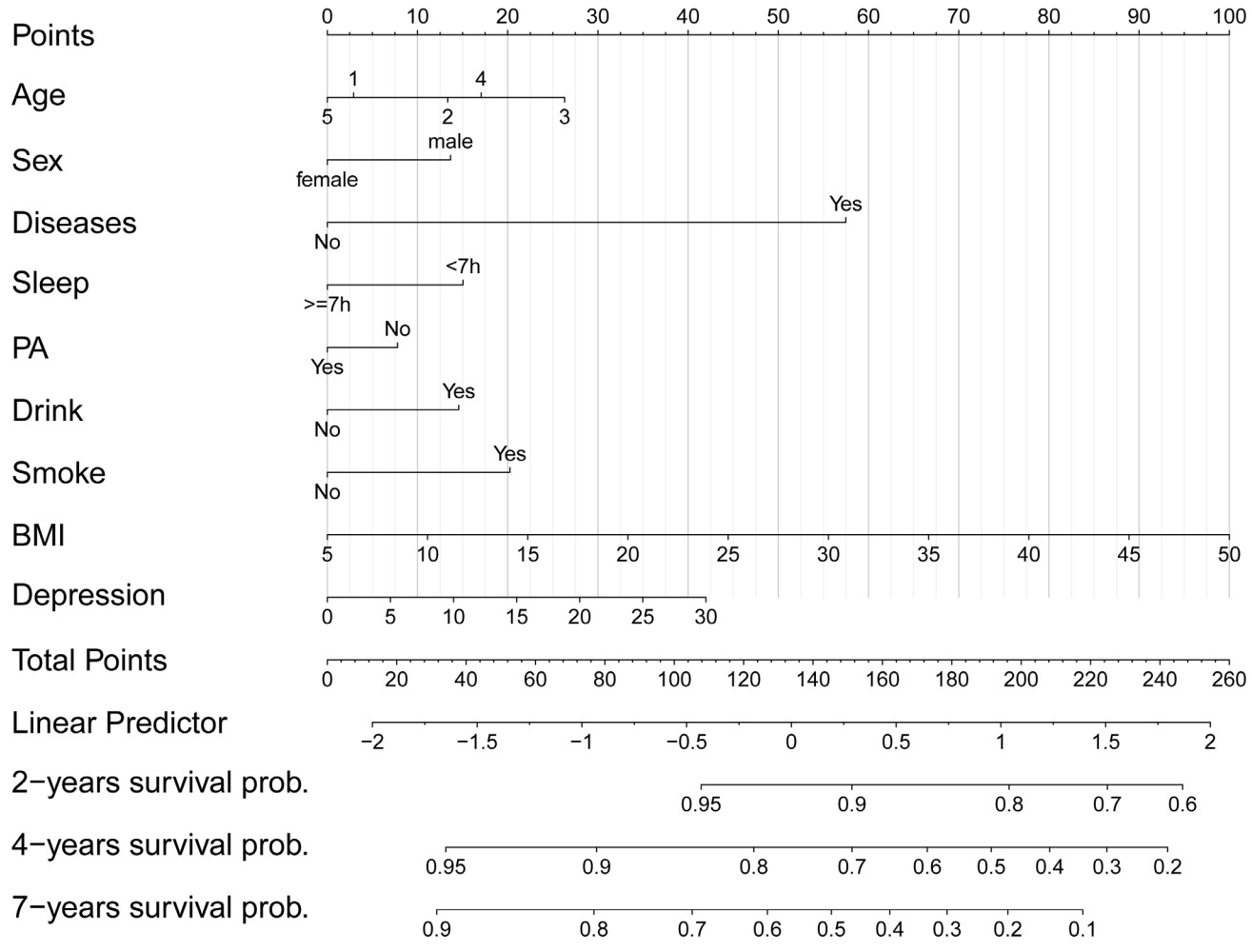

**Figure 2** The nomogram for multimorbidity risk prediction. Note: Draw a line perpendicular from the corresponding axis of each risk factor until it reaches the line labelled 'POINTS'. Sum up the number of points for all risk factors then draw a line descending from the labelled 'TOTAL POINTS' until it intercepts each of the survival axes to determine 2-year, 4-year and 7-year survival probabilities, multimorbidity probability=1−survival probability. BMI, body mass index; PA, physical activity.

### Development and validation of an multimorbidity predicting nomogram

The PI was calculated based on the HR associated with the identified risk factors for multimorbidity. The nomogram was constructed using these results, with the BMI variable assigned a total scale of 100 and a range of 5–50. The risk score for BMI was determined to be 2.2. The risk scores for the other risk factors of multimorbidity were calculated proportionally to the β coefficient of BMI. So the PI=(0.3×I(age-45))+(14.3×I(male))+(56.9× I(chronic disease))+(14.8×I(1-sleep ≥7 hours))+(6.4×I(1-regular PA))+(13.7×I(drink))+(19.1×I(smoke))+(2.2×BMI)+(1.4×depression), where I() denotes the indicator function equal to 1 if the condition in parenthesis is met and 0 otherwise, except age. Based on these findings, a nomogram was configured (figure 2).

The resulting nomogram was internally validated using the bootstrap validation method, and it demonstrated good accuracy in estimating the risk of multimorbidity, with a bootstrap-corrected C-index of 0.70 (95% CI 0.69 to 0.71, p=0.006) in the derivation set. Calibration plots also indicated good agreement between the risk estimation by the nomogram and the diagnosis of doctors, as depicted in figure 2. When the estimates from the derivation set were applied to the validation set, a similar bootstrap-corrected C-index of 0.71 (95% CI 0.70 to 0.73, p=0.008) was obtained, along with a well-calibrated risk estimation curve (figure 3).

### DISCUSSION

In this study, we observed a prevalence of multimorbidity of 46.2% among middle-aged and older adults. The prevalence of multimorbidity was higher among individuals with one chronic disease (62.8%) compared with those without chronic diseases (30.8%) at baseline. This result is consistent with the probability range of multimorbidity reported in a systematic study of the elderly.[8] Although there were some differences in the estimation of multimorbidity compared with other studies, such as the number of included chronic diseases,[36 37] and the methods used to collect information.[38] Our results confirmed the

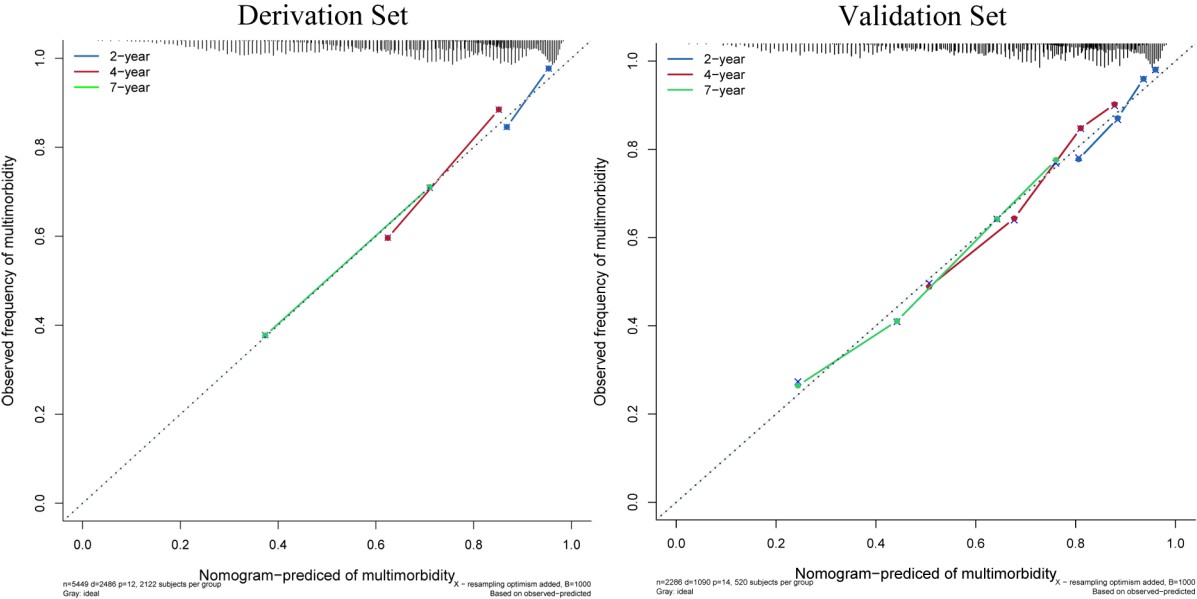

**Figure 3** The predictive performance of the nomogram in estimating the risk of multimorbidity.

significant impact of multimorbidity on middle-aged and older adults, particularly among those with one chronic disease.

Our findings provide sufficient evidence for the association between older age and multimorbidity, which is consistent with similar trends observed in countries such as Singapore,[39] Ireland and Scotland.[27] The prevalence of multimorbidity among women is higher than men. Additionally, we found a higher prevalence of multimorbidity among women compared with men, which is supported by studies from various countries indicating that older men have a lower risk of multimorbidity than their female counterparts.[27 38] However, a study by Lian found that the onset of multimorbidity occurs at an earlier age in men than in women.[39] This is also a new idea to understand the sex-differ in multimorbidity. Future research could explore the prevalence and different combinations of chronic conditions in people with multimorbidity across various age and sex groups.

Existing studies have established a link between short sleep duration and multimorbidity. Experimental evidence confirms the deleterious effects of sleep deprivation on endocrine, immune, neurovitality and inflammatory pathways.[40] For instance, Maria Ruiz-Castell et al found an association between short sleep duration and the number of chronic conditions.[41] Helbig et al also observed a significant positive relationship between short sleep duration and multimorbidity among women.[42] High-risk lifestyles, such as smoking, excessive alcohol consumption, poor diet, physical inactivity and unhealthy body shape, have also been confirmed as contributing factors to multimorbidity.[43] Smoking and excessive alcohol consumption remain leading risk factors for early death and disability globally.[44 45] In our study, we found that smoking and drinking increased the risk of multimorbidity, with the

highest risk index among all unhealthy behaviours. The expansion of tobacco and alcohol control measures remains a significant public health priority worldwide. Mika conducted an observational study using data from two Finnish cohort studies comprising 614014 adults, and the results showed that obesity is a significant factor in multimorbidity.[46] Similarly, our study found that higher BMI was associated with an increased risk of multimorbidity among middle-aged and older adults. There exists a bidirectional association between depression and multimorbidity.[16 47] Depression increases the risk of multimorbidity, while having multimorbidity also raises the risk of depression. Our study found the risk of multimorbidity for middle-aged and older adults with higher depression scores.

Based on our results, we developed a user-friendly nomogram model for predicting the risk of multimorbidity. One of the most appealing aspects of our nomogram model is its home applicability and ease of use by individuals. For example, a 50-year-old married man with primary education, one chronic disease, a history of smoking and excessive alcohol consumption, irregular physical activity, 8 hours of sleep, a depression score of 16 and a BMI of 24.9 would have a total risk score of 219.3 points. This corresponds to a 2-year, 4-year and 7-year probability of multimorbidity of 32%, 71% and <10%, respectively. Based on the calculated results, individuals can develop self-management strategies to reduce their risk of multimorbidity.

### Limitation

Our study had several limitations. First, the model was constructed based on behavioural and household-level risk factors, limiting its applicability to clinical prediction. Additionally, in this study, there is a large proportion of

missing data for some important variables, so we have used a culling method to deal with the missing values. As a result, this may lead to sample bias and the extrapolation of the model needs to be careful. We also need to validate the model using external data.

## Conclusions

This study confirms the severity of multimorbidity among middle-aged and older adults, particularly among those who already have one chronic disease. Age showed a significant correlation with multimorbidity, and the prevalence of multimorbidity was higher in women compared with men. In addition, insufficient sleep, smoking, drinking, obesity and depressive symptoms were also associated with multimorbidity. Based on these findings, we developed a user-friendly nomogram model to predict the risk of multimorbidity in middle-aged and older adults. Our research not only builds on the existing body of knowledge but also introduces a novel and comprehensive approach to assessing multimorbidity risk, which is of significant clinical and public health relevance. The multivariable prediction model provides valuable tools for healthcare professionals to manage multimorbidity.

**Author affiliations**
[1]Department of Health Management, Shunde Hospital, Southern Medical University (The First People's Hospital of Shunde, Foshan), Foshan, China
[2]School of Health Management, Southern Medical University, Guangzhou, China
[3]Key Laboratory of Philosophy and Social Sciences of Guangdong Higher Education Institutions for Collaborative Innovation of Health Management Policy and Precision Health Service, Guangzhou, China

**Contributors** ZX, ZCC and TF conceived the study, as well as supervised the study. XZ was major contributor in writing the manuscript. XBL, XSJ and LXR analysed and interpreted the patient data. CYM, SL and LXY edited and contributed content to the final draft. All authors read and approved the final manuscript.

**Funding** This study was funded by the Guangdong Basic and Applied Basic Research Foundation (No.2022A1515110295, No.2022A1515011591), National Nature Science Foundation of China (No.72274091), China Postdoctoral Science Foundation (No. 2022M721539), Guangdong Philosophy and Social Science Foundation (No. GD23CGL06).

**Competing interests** None declared.

**Patient and public involvement** Patients and/or the public were not involved in the design, or conduct, or reporting, or dissemination plans of this research.

**Patient consent for publication** Not applicable.

**Ethics approval** The Biomedical Ethics Review Committee of Peking University approved CHARLS, and all participants were required to provide written informed consent. The ethical approval number was IRB00001052-11015.

**Provenance and peer review** Not commissioned; externally peer reviewed.

**Data availability statement** Data are available upon reasonable request. The data sets analysed during the current study are available from the corresponding author on reasonable request.

**ORCID iDs**
Lei Shi http://orcid.org/0000-0002-8924-0734
Chichen Zhang http://orcid.org/0000-0003-1095-9939

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
