## [Reviewer comments · BMJ Open]

ARTICLE DETAILS

TITLE (PROVISIONAL)	Development and validation of a multimorbidity risk prediction nomogram among Chinese middle-aged and older adults: a retrospective cohort study
AUTHORS	Zheng, Xiao; Xue, Benli; Xiao, Shujuan; Li, Xinru; Chen, Yimin; Shi, Lei; Liang, Xiaoyan; Tian, Feng; Zhang, Chichen

VERSION 1 – REVIEW

REVIEWER	Saito, Yoshiyuki The University of Tokyo, Department of Health Economics & Outcomes Research
REVIEW RETURNED	03-Aug-2023

GENERAL COMMENTS	Thank you for affording me the opportunity to review such a significant study. I find your endeavor to predict multimorbidity using a simple model to be of great importance. However, I perceived some issues concerning the research design and statistical methods etc. I would recommend defining multimorbidity and health behaviors in accordance with prior research. If not, the reasons for this divergence should be explicitly stated. Additionally, when creating a predictive model in epidemiological papers, I suggest you refer to guidelines like the TRIPOD Checklist to guide your research. It would also be beneficial to discuss potential biases or limitations inherent to your chosen study population in the limitations of your study.
---

REVIEWER	Munyombwe, T University of Leeds, Leeds Institute of Cardiovascular and Metabolic Medicine
REVIEW RETURNED	30-Aug-2023

GENERAL COMMENTS	The researchers developed a nomogram to predict the risk of multimorbidity among middle-aged and older adults. Data from the Chinese longitudinal health longevity survey with n=7735 participants was used. A multivariable cox proportional hazards model was used. The study concluded that the nomogram that was developed can facilitate early identification of high-risk individuals of multimorbidity to provides personalized preventive and intervention. I have minor comments as follows: Title The title of the manuscript is misleading, since the main aim of this study was to develop a risk prediction model in form of a nomogram, this should be reflected in the title. This study should be identified as developing and validating a multivariable
---

	prediction model for multimorbidity in middle and older age adults: Nomogram approach. (see TRIPOD guidelines) Abstract  -The hazard ratio 95% CI for age in the abstract is non-significant as these includes 1. (1.01, 95%CI 1.00-1.01). - Consider reporting the model performance measures discrimination and calibration and corresponding 95%CI in the abstract. -I don't see the relevance of including the equation of the prediction model in the abstract. Details of the prediction model can be reported in the manuscript. Background The risk factors of multimorbidity identified by the authors are well established. May the authors provide a rationale for developing their multivariable prediction model, are there no other existing models that they could have updated ? Provide references of existing models and critique them to justify this current study. Methods Outcome  -Clearly define the outcome that is predicted by the prediction model, you stated the outcome as development of multimorbidity , since you used a survival model, was the outcome time to multimorbidity and is it in years ? -If the outcome was presence of multimorbidity (Binary) the authors should have used a multivariable logistic regression model. Predictors  - You reported that eleven meaningful variables were included in the multivariable model, state the variables in the methods section and how these were handled in the model. At what point were these variables measured, is it at baseline ? Variable selection A better approach is to use backward selection to select the variables to include in the model (see recommendations from Richard Riley) Model performance  -Report the confidence intervals for measures of model performance Discussion  -You used a complete case analysis excluding participants with missing data, highlight the limitation of this approach in the discussion e.g non representation. -This model was internally validated, more research is required for external validation of the model before use.
--	--

VERSION 1 – AUTHOR RESPONSE

Dear Reviewer 1:

Thank you very much for reviewing our article. Your advice and guidance are greatly appreciated. Following thorough consideration and discussions, we have implemented the necessary revisions in the article and have tracked these changes in red within the manuscript. In the following sections, we provide a point-by-point response to each of your valuable comments and suggestions.

Q1: I would recommend defining multimorbidity and health behaviors in accordance with prior research. If not, the reasons for this divergence should be explicitly stated.

Answer: Multimorbidity, commonly defined as the co-occurrence of two or more chronic conditions. But the disease patterns of multimorbidity is undefined. A systematic review of 14 studies conducted in Europe and the US has identified 97 disease patterns of multimorbidity. Despite heterogeneity in multimorbidity patterns across studies due to the diversity of the study population, diseases included, and statistical methods, three common multimorbidity patterns were observed, including cardiovascular and metabolic diseases, mental health problems, and musculoskeletal disorders. A study of prof. Liming Li is the first study in China to comprehensively examine the association of both the number of chronic diseases and the patterns of multimorbidity, using data from the China Kadoorie Biobank (CKB), including 512,725 participants aged 30 to 79 years were recruited from ten (five urban and five rural) study areas across China during 2004 to 2008. In the study, the multimorbidity includes 15 chronic diseases, namely, hypertension, diabetes, coronary heart disease, stroke or transient ischemic attack, tuberculosis, asthma, chronic obstructive pulmonary disease (COPD), gallstone diseases, peptic ulcer, cirrhosis/chronic hepatitis, chronic kidney disease, cancer, neurasthenia, psychiatric disorder, and rheumatoid arthritis. The definition of multimorbidity among Chinese in most studies is consistent with this study. In this study, we have also adopted the same definition of multimorbidity. We cannot include all chronic diseases in the study, but we selected 14 chronic diseases with high prevalence in the Chinese population, which we believe can be used to evaluate multimorbidity in the Chinese population.

This study includes four well-known healthy lifestyle factors, which are the four cornerstones of a healthy lifestyle proposed by the World Health Organization in 1992, namely physical activity (PA), smoking, alcohol consumption, and dietary behavior. In addition, we also included the health behaviors found in previous studies, including sleep and social activity.

Q2: Additionally, when creating a predictive model in epidemiological papers, I suggest you refer to guidelines like the TRIPOD Checklist to guide your research.

Answer: Thanks for your suggestion. We have made revisions to the article based on the TRIPOD guidelines and the comments of the reviewers.

Q3: It would also be beneficial to discuss potential biases or limitations inherent to your chosen study population in the limitations of your study.

Answer: In this study, there is a large proportion of missing data for some important variables, so we have used a culling method to deal with the missing values. As a result, this may lead to sample bias and the extrapolation of the model needs to be careful. We also need to validate the model using external data. Please see Limitation section on page 18.

Dear Reviewer 2:

Thank you for your thorough review of our manuscript and for providing valuable feedback. We appreciate the time and effort you have dedicated to evaluating our work. In response to your comments, we have made the following revisions to improve the quality of the manuscript.

Q1: Title

The title of the manuscript is misleading, since the main aim of this study was to develop a risk prediction model in form of a nomogram, this should be reflected in the title. This study should be identified as developing and validating a multivariable prediction model for multimorbidity in middle and older age adults: Nomogram approach. (see TRIPOD guidelines)

Answer: Thank you for your suggestion. We have changed the title to "Development and Validation of a multivariable prediction model for Multimorbidity Risk in middle-aged and older adults: A Nomogram approach".

Q2: Abstract

-The hazard ratio 95% CI for age in the abstract is non-significant as these includes 1. (1.01, 95%CI 1.00-1.01).

Answer: Thank you for your suggestion. We divided the age into 5 levels (<50 y, 50-59 y, 60-69 y, 70-79 y, >=80 y) and included them in the model. The results are as follows:

In comparison to individuals under the age of 50, those in the age groups of 50-59 (HR 1.18, 95%CI 1.04-1.33) , 60-69 (HR 1.44, 95%CI 1.27-1.64), and 70-79 (HR 1.23, 95%CI 1.04-1.45) exhibit a higher risk of multiple chronic diseases. Please refer to Table 2 on pages 13-14.

Q3: Consider reporting the model performance measures discrimination and calibration and corresponding 95%CI in the abstract.

-I don't see the relevance of including the equation of the prediction model in the abstract. Details of the prediction model can be reported in the manuscript.

Answer: Thank you for your suggestion. We have added the C-index and their 95%CI in the abstract. Please see in page 2.

"The C-index of nomogram models for derivation and validation sets were 0.70 (95%CI 0.69-0.71, P=0.006) and 0.71 (95%CI 0.70-0.73, P=0.008), respectively."

Q4: Background

The risk factors of multimorbidity identified by the authors are well established.

May the authors provide a rationale for developing their multivariable prediction model, are there no other existing models that they could have updated? Provide references of existing models and critique them to justify this current study.

Answer: Thank you for your valuable suggestions. In revising our background section, we have incorporated your advice and restructured the background to convey the necessity and innovation of our research more clearly:

Although age is undeniably a well-established risk factor for multimorbidity(7), the multifaceted nature of this phenomenon demands a more comprehensive understanding that goes beyond age-related associations. Existing research has indeed confirmed the higher prevalence of multimorbidity among older adults, with systematic reviews reporting rates ranging from 55% to 98% in the elderly population(8). Additionally, demographic factors such as older age, female gender, and lower socioeconomic status have consistently been associated with an increased risk of multimorbidity. However, while these factors provide valuable insights, they represent only a fraction of the complex web of variables contributing to multimorbidity.

Previous studies have explored associations between sociodemographic factors, physical characteristics, and social networks with multimorbidity(9, 10). Previous studies have explored associations between sociodemographic factors, physical characteristics, and social networks with multimorbidity. For example, numerous studies have linked poor sleep quality to an elevated risk of multimorbidity(11-14). Furthermore, the impact of body mass index (BMI) and smoking on multimorbidity prevalence underscores the intricate connections among these factors(15). Additionally, the well-documented relationship between depression and common chronic diseases has been established(16, 17), with longitudinal cohort studies demonstrating bidirectional associations between depression and multimorbidity(18).

Despite the substantial body of evidence regarding the associations among sociodemographic factors, social networks, lifestyle factors, depression, and the risk of developing multimorbidity, there remains a notable gap in the literature. This gap pertains to the absence of a comprehensive multivariable prediction model that integrates all these factors, providing a holistic assessment of multimorbidity risk. Our study seeks to address this gap by developing and validating a novel risk assessment model that encompasses a broad spectrum of variables, including those mentioned above. Our aim is to equip individuals with a more accurate and personalized estimate of their risk of developing multimorbidity, contributing to a deeper understanding of this multifaceted health issue.

Wider determinants of health (WDHs) encompass a multitude of social, economic, political, and environmental factors that exert influence on health outcomes across an individual's lifespan. This influential model of health determinants places constitutional factors such as sex, age, and genetics at its core, surrounded by concentric layers that encompass individual lifestyle factors, followed by the broader determinants(19). While the core attributes remain relatively fixed, the determinants become more modifiable as the layers extend outward. Existing research has identified that individual lifestyle factors significantly contribute to multimorbidity among older adults. Chudasama also found that adopting a healthier lifestyle was associated with longer life expectancy for middle-aged adults, regardless of the presence of multimorbidity.(4).

In the context of the WDHs framework, individual behaviors constitute the innermost layer, presenting opportunities for modification, particularly through self-health management(20-22). The accurate assessment of one's risk of developing multimorbidity and the identification of potential risk factors represent critical initial steps in the journey of self-management. Therefore, the development of a user-friendly tool to assist individuals in estimating their risk of multimorbidity is of paramount significance.

We believe that our research not only builds upon the existing body of knowledge but also introduces a novel and comprehensive approach to assessing multimorbidity risk, which is of significant clinical and public health relevance. We hope that these revisions clarify the rationale for our study and highlight the innovative aspects that distinguish it from existing research.

Please refer to the Background section on pages 3-5.

Q5: Outcome

-Clearly define the outcome that is predicted by the prediction model, you stated the outcome as development of multimorbidity, since you used a survival model, was the outcome time to multimorbidity and is it in years?

-If the outcome was presence of multimorbidity (Binary) the authors should have used a multivariable logistic regression model.

Answer: Thank you for your suggestion. The outcome was the time to multimorbidity, we have added it in the Measurements section (Multimorbidity). Please see in page 7.

Q6: Predictors

- You reported that eleven meaningful variables were included in the multivariable model, state the variables in the methods section and how these were handled in the model. At what point were these variables measures, is it at baseline?

Answer: Thank you for your suggestion. A detailed introduction to the predictive variables has been added to the methods section.

The covariates, including demographic characteristics and modifiable lifestyles factors were gathered by baseline questionnaire.

Please see in pages 6-8.

Q7: Variable selection

A better approach is to use backward selection to select the variables to include in the model (see recommendations from Richard Riley)

Answer: We sincerely appreciate your valuable suggestion. In accordance with your advice, we have employed a backward selection approach to determine the variables included in the model. As a result, the multivariable model now comprises 9 significant variables, with education and marital status being excluded. For detailed information, please refer to Table 2 on pages 13-14.

"The multivariable Cox proportional hazard models incorporated eleven significant variables identified through univariate analysis. Variable selection was performed using the backward selection method. In comparison to individuals under the age of 50, those in the age groups of 50-59 (HR 1.18, 95%CI

1.04-1.33), 60-69 (HR 1.44, 95%CI 1.27-1.64), and 70-79 (HR 1.23, 95%CI 1.04-1.45) exhibit a higher risk of multimorbidity. Furthermore, being female (HR 1.23, 95%CI 1.10-1.36), having a history of chronic disease (HR 2.59, 95%CI 2.38-2.82), getting at least 7 hours of sleep (HR 0.78, 95%CI 0.72-0.85), engaging in regular PA (HR 0.88, 95%CI 0.81-0.95), drinking (HR 1.27 95%CI 1.16-1.39), smoking (HR 1.40, 95%CI 1.26-1.53), higher BMI (HR 1.04, 95%CI 1.03-1.05) and increased depression scores (HR 1.02, 95%CI 1.01-1.03) were all found to be associated with multimorbidity in derivation set (Table 2). ”

Q8: Model performance

-Report the confidence intervals for measures of model performance

Answer: Thank you for your suggestion. We have added the confidence intervals for measures of model performance. Please see in page 15.

“The resulting nomogram was internally validated using the bootstrap validation method, and it demonstrated good accuracy in estimating the risk of multimorbidity, with a bootstrap-corrected C-index of 0.70 (95%CI 0.69-0.71, P=0.006) in the derivation set. Calibration plots also indicated good agreement between the risk estimation by the nomogram and the diagnosis of doctors, as depicted in Figure 2. When the estimates from the derivation set were applied to the validation set, a similar bootstrap-corrected C-index of 0.71 (95%CI 0.70-0.73, P=0.008) was obtained, along with a well-calibrated risk estimation curve (Figure 3).”

Q9: Discussion

-You used a complete case analysis excluding participants with missing data, highlight the limitation of this approach in the discussion e.g non representation.

-This model was internally validated, more research is required for external validation of the model before use.

Answer: In this study, there is a large proportion of missing data for some important variables, so we have used a culling method to deal with the missing values. As a result, this may lead to sample bias and the extrapolation of the model needs to be careful. We also need to validate the model using external data. Please refer to the Limitation section on page 18.

VERSION 2 – REVIEW

REVIEWER	Munyombwe, T University of Leeds, Leeds Institute of Cardiovascular and Metabolic Medicine
REVIEW RETURNED	13-Oct-2023
GENERAL COMMENTS	The authors have addressed my comments satisfactorily and I have no further suggestions.

VERSION 2 – AUTHOR RESPONSE

Comments to the Author:

The authors have addressed my comments satisfactorily and I have no further suggestions.

Answer: We would like to thank you again for taking the time to review our manuscript.